# Individualised Nutritional Care for Disease-Related Malnutrition: Improving Outcomes by Focusing on What Matters to Patients

**DOI:** 10.3390/nu14173534

**Published:** 2022-08-27

**Authors:** Anne Holdoway, Fionna Page, Judy Bauer, Nicola Dervan, Andrea B. Maier

**Affiliations:** 1Bath Clinic, Circle Health Group, Bath BA2 7BR, UK; 2First Page Nutrition Ltd., Chippenham SN15 5HS, UK; 3Department of Nutrition, Dietetics & Food, Monash University, Clayton, VIC 3168, Australia; 4School of Public Health, Physiotherapy and Sports Science, University College Dublin, DO4 V1W8 Dublin, Ireland; 5Institute of Food and Health, University College Dublin, DO4 V1W8 Dublin, Ireland; 6Department of Human Movement Sciences, @AgeAmsterdam, Vrije Universiteit Amsterdam, Amsterdam Movement Sciences, 1081 BT Amsterdam, The Netherlands; 7Department of Medicine and Aged Care, @AgeMelbourne, The Royal Melbourne Hospital, The University of Melbourne, Parkville, VIC 3050, Australia; 8Healthy Longevity Translational Research Program, Yong Loo Lin School of Medicine, National University of Singapore, Singapore 117456, Singapore; 9Centre for Healthy Longevity, @AgeSingapore, National University Health System, Singapore 119074, Singapore

**Keywords:** malnutrition, nutritional support, patient preferences, guidelines

## Abstract

Delivering care that meets patients’ preferences, needs and values, and that is safe and effective is key to good-quality healthcare. Disease-related malnutrition (DRM) has profound effects on patients and families, but often what matters to patients is not captured in the research, where the focus is often on measuring the adverse clinical and economic consequences of DRM. Differences in the terminology used to describe care that meets patients’ preferences, needs and values confounds the problem. Individualised nutritional care (INC) is nutritional care that is tailored to a patient’s specific needs, preferences, values and goals. Four key pillars underpin INC: what matters to patients, shared decision making, evidence informed multi-modal nutritional care and effective monitoring of outcomes. Although INC is incorporated in nutrition guidelines and studies of oral nutritional intervention for DRM in adults, the descriptions and the degree to which it is included varies. Studies in specific patient groups show that INC improves health outcomes. The nutrition care process (NCP) offers a practical model to help healthcare professionals individualise nutritional care. The model can be used by all healthcare disciplines across all healthcare settings. Interdisciplinary team approaches provide nutritional care that delivers on what matters to patients, without increased resources and can be adapted to include INC. This review is of relevance to all involved in the design, delivery and evaluation of nutritional care for all patients, regardless of whether they need first-line nutritional care or complex, highly specialised nutritional care.

## 1. Introduction

The World Health Organisation (WHO) defines quality in healthcare as effective, safe and people-centred, highlighting the need to provide care that meets individual preferences, needs and values, whilst also being timely, equitable, integrated and efficient [1]. People-centred care is care that adopts the perspectives of individuals, carers, families and communities and is organised around the needs of people rather than specific diseases. It is wider-reaching than patient-centred care since it includes clinical encounters, consideration of the health of people in their communities and their role in shaping health policy and services [2]. 

From a patient’s perspective, the key elements of patient-centred care can be summarised under the following domains: (i) meets patients’ preferences, (ii) provides information that is understandable by the individual and carers, (iii) is accessible, (iv) delivers emotional support where needed, (v) addresses the needs of family and friends, (vi) provides continuity as individuals transition across care settings, (vii) addresses physical comfort and (viii) coordination of care [3]. 

Information is often lacking on how healthcare services impact people beyond measures of complications, hospital re-admissions and death. The Organisation for Economic Co-operation and Development (OECD) Health Care Quality and Outcomes programme aims to address this by driving international policy development that focuses on evaluating outcomes from the perspective of the people served, i.e., measuring what matters to patients [4]. 

Disease-related malnutrition (DRM) can have profound effects on an individual and their families, not only due to the poor physical function that may arise from weight loss, but also from the disruption of daily routines and social interactions, and anxiety around eating and drinking. Nutritional care is a therapy and not just supportive care. Studies investigating the effectiveness of nutritional care on DRM have focused on anthropometric data, measuring food and nutrient intake against estimated or measured nutritional requirements, and healthcare-related outcomes, such as hospital length of stay, complications and hospital readmissions, with a limited number reporting the effects of nutritional care on quality of life (QOL) [5]. Considering the broad domains of patient-centred care, studies that investigate the effectiveness of nutritional care need to go beyond the realms of these measures if we are to measure the impact of nutritional care strategies and evaluate their effectiveness in meeting patient’s needs, preferences, values and goals.

With an ageing population and co-existing morbidities being commonplace [6], nutritional care has had to evolve to deliver care that addresses not only one medical condition, but the nutritional and dietary demands of several medical conditions. The combination of conditions, including DRM, may result in competing priorities in nutritional care further driving the need for an individualised approach to deliver the most appropriate care according to the patient’s needs at that point in time. Much can be learned from the holistic approach taken during palliative care where the essential practices for primary palliative care cover physical care; psychological, emotional and spiritual care; care planning; coordination and communication, and takes account of the values and preferences of the patient and their families [7]. In addition, as the benefits associated with delivering patient-centred care extend beyond the patient and can contribute to greater job satisfaction among healthcare professionals [8], organisations may gain benefit by delivering care that is more patient-centred.

The aim of this article is to review how individualised nutritional care (INC) that meets patients’ needs, preferences, values and goals is a key part of patient-centred care. This review considers the merit in adopting an individualised approach to deliver patient-centred care in the management of DRM. After considering the evidence-base supporting an individualised approach to nutritional care and the incorporation of the concept in relevant nutrition guidelines, the review explores how INC can be delivered in clinical practice across healthcare settings, by all healthcare disciplines. How such an approach might assist healthcare professionals deliver care that achieves outcomes that matter to patients is also considered. 

## 2. Patients at the Centre of Their Care

### 2.1. Terminology

The terminology used to describe patient-centred care may vary across healthcare systems and within healthcare practice. Despite variations, the need to provide care that meets patients’ individual preferences, needs and values is a consistent theme. Table 1 defines these terms and also defines goals, since goal setting, during intervention planning, is a key part of tailoring care to the individual. The co-creation of personally relevant goals and advice aligned with patients’ personal preferences are key factors to motivate engagement in care and support self-management [9,10]. Terms such as targeted, stratified, tailored, personalised, precision, customised and individualised nutrition are commonly used, but are often used interchangeably and remain ill-defined.

Vague or imprecise terms leave meaning open to interpretation, making comparisons between studies and implementation in practice difficult. In nutritional care, the term ‘patient-centred care’ lacks a single, clear definition leading to a variability in how patient-centred care is conducted [16]. Key themes of patient-centred nutritional care include establishing positive dietitian–patient relationships, displaying humanistic behaviours (behaviours or characteristics perceived as helpful and motivating to patients), using effective communication, individualising and adapting care and redistributing power to patients [17]. Patients’ perceptions and experience of patient-centred care in dietetic consultations are similar, in that patients want individualised care, caring relationships, involvement in care processes and to take control of their own health [18]. There is perhaps a need to work towards a more precise definition of nutritional care that is patient-centred, resonates with healthcare professionals and patients and encompasses these themes. 

### 2.2. Defining Individualised Nutritional Care 

It is proposed that INC provides a useful description to reflect nutritional care that is tailored to a patient’s specific needs, preferences and goals. Underpinning the successful delivery of INC are four key pillars: (i) exploring carefully what matters to patients; (ii) involving patients as equal partners in decisions about their care (shared decision making); (iii) employing multiple methods of nutritional care (multi-modal) as informed by evidence (where available), based on thorough nutritional assessment and diagnosis and (iv) timely monitoring and evaluation to measure the effectiveness of nutritional care towards co-created patient goals. Fostering ongoing relationships, taking active measures to ensure continuity of nutritional care across healthcare settings and encouraging patient self-monitoring are essential to success, the aim being to optimise nutritional care to support adherence and improve outcomes that matter to patients (Figure 1). 

Multi-modal nutritional care refers to the use of multiple methods of nutritional care that might be employed by one or more members of the healthcare team. In managing DRM, it can include a single or a combination of nutritional care interventions, e.g., dietary counselling, dietary modifications, oral nutritional supplements (ONS), enteral nutrition (EN) or parenteral nutrition (PN), and the use of specific targeted nutrients, e.g., vitamin D; protein; omega-3 fatty acids and other interventions, such as exercise and psychological support [19]. Nutritional care includes nutritional assessment and diagnosis, which informs the nutritional intervention(s), which, in turn, is monitored and evaluated. 

Nutritional care should be evidence-based and individualised across all such nutritional care activities, i.e., assessment, diagnosis, intervention, monitoring and evaluation, at all time points in a patient’s healthcare journey and across all healthcare settings. INC is relevant to all methods of nutritional intervention. Methods or a combination of methods of nutritional intervention should be used that reflect the evidence base and best meet the patient’s needs, preferences and goals. Monitoring should establish progress with regard to the nutrition diagnosis and allow nutritional care to be adapted based on measuring outcomes that matter to patients, combined with outcome measures required by the care setting or provider. Selecting outcome measures or indicators that are meaningful to patients and using lay terms for goal setting may help patients self-monitor and engage in nutritional care.

For example, for patients requiring enteral tube feeding, particularly in the community setting, INC includes discussion of the type of tube, method of administration (pump, bolus, intermittent), feeding regimen and the types of feed. Counselling patients on options available can empower them to make choices and enable them to alert healthcare professionals when issues arise that could be managed by altering aspects of feed choice, for example, the use of fibre-containing feeds to manage constipation or diarrhoea, or a combination of methods of administration to fit with clinical and lifestyle needs. For patients on ONS, selection of formats, flavours, textures, volume and nutritional content based on individual patient needs, preferences and goals could play a key role in supporting adherence. 

## 3. The Inclusion of Individualised Nutritional Care in the Delivery of Nutritional Care to Manage Disease-Related Malnutrition

### 3.1. Individualised Nutritional Care in Nutrition Guidelines

To evaluate the incorporation of an individualised approach to nutritional care in the management of DRM, a review was undertaken of nutrition guidelines focused on the prevention and management of DRM (general provision of nutrition support, intensive care, COVID-19, cancer and older people) published by internationally recognised professional organisations from 2017 to 2022. Each guideline was searched to identify information, recommendations or standards relating to INC as defined in Figure 1. Table 2 lists those guidelines that mentioned elements of INC as defined in Figure 1 and provides a summary of the relevant key recommendations and statements. Refer to online Appendix A for a table that includes the full details of the information relating to INC in the original publications. 

The term INC is commonly used in medical nutrition practice as evidenced by its use within nutrition guidelines. However, there is a range of ways in which INC is represented within guidelines including: consideration of patient needs, preferences and values [20,21,22,25,35,36]; consideration of patient goals in cancer patients [22,33] and in-patients receiving EN or PN, although it was not clear if these were patient goals or goals as identified by a HCP [25,26]; individual assessment or adjustment of nutritional requirements (energy and protein) [21,27,30,35] and INC plans or nutrition intervention [21,23,26,29,30]. In certain topics, such as the nutritional care of older people and patients with cancer, the descriptions of INC are broader and incorporate inter- or multidisciplinary, multi-modal approaches where the patient’s personal goals are also considered [32,33,34,35,36].

Singer et al. (2019) in the ESPEN guideline on clinical nutrition in the intensive care unit acknowledge the value of recommendations whilst highlighting that, due to the large heterogeneity of the ICU population, they can only serve as a basis to support decisions made for each patient on an individual basis [28]. Other guidelines also acknowledge that recommendations are general, but they should not replace clinical judgments for individual patients [24,26]. The challenge remains on how to implement guidelines whilst making sure care is individualised to patients’ needs, preferences, values and goals. There may be a need for guidelines to include not only recommendations on the need for INC, but also detail on how to deliver INC. 

It therefore appears that although INC is embedded in evidence-based guidelines, there is variation in the degree to which it is described. A recent scoping review explored the delivery of dietetic patient-centred care within subacute rehabilitation units (six studies in two countries). Although elements of patient-centred care, such as individualising interventions, continuity of care and ongoing follow-up, were consistently mentioned, other essential components of patient-centred care, such as understanding and acknowledging what is important to the patient, team collaboration and shared decision making with the patient and their family, were lacking in many studies [37]. It may be that patient-centred care is not fully applied, adequately described or reported [37]. 

### 3.2. Inclusion of Individualised Nutritional Care in Studies of Oral Nutritional Intervention

In 2021, Baldwin et al. published an updated Cochrane review examining the evidence that oral nutritional intervention for DRM in adults improves mortality, morbidity, weight, anthropometry, dietary intake and QOL [5]. Comparisons were made between various combinations of dietary advice, with and without ONS and no dietary advice. Overall, there was no difference in mortality for any of the comparisons; however, positive changes were observed for dietary intake, body weight, fat-free mass and QOL. Additionally, Baldwin et al. reported that dietary advice plus ONS may lead to fewer complications and a shorter length of hospital stay after three months in adult patients with DRM [5]. 

Table 3 shows the number of studies included by Baldwin et al. where the nutrition intervention for DRM in adults was described as individualised or individualisation was not mentioned in the description of the intervention. Figure 2 shows there has been a higher number of studies reporting aspects of individualisation over the last 20 years. 

Baldwin et al. observed a positive effect of oral nutritional intervention on QOL, and 18 of the 23 studies included in the analyses of change in global QOL were described as individualised [5]. However, how nutritional care interventions in studies are individualised is variably described, not consistent or not described in detail and may be variably interpreted or applied, highlighting the lack of a consensus definition. Descriptions may be limited to individualising nutritional care to usual dietary intake or preferences or to reach individual dietary intake goals or estimated requirements. Aspects of patient-centred care, such as patients’ values, needs, preferences and goals, and shared decision making are often either not explicitly included as elements of care interventions or not described in study reports [37], hindering comparisons. Thus, although comparisons 4 and 5 in Table 3 included ≥90% of studies that indicated some element of individualisation of nutritional care, it cannot be assumed that all these studies employed a fully individualised approach as outlined in Figure 1. 

It should be recognised that the Baldwin review does not represent the entirety of the evidence base for nutritional care; rather, the authors focused on dietary advice with or without ONS for the management of DRM in adults. Studies of EN, PN, disease-specific ONS and multi-modal interventions in which the nutritional intervention was combined with exercise or other interventions were excluded. Further limitations of the review were that it included patients across many different conditions, and studies published in the last three years (some of which describe the nutritional intervention as individualised) were not included. 

To address some of these limitations, a systematic review and meta-analysis were undertaken to investigate the effectiveness of dietary counselling (1 to ≥3 sessions, mostly dietitian-led) with or without ONS in at-risk or malnourished hospitalised patients (16 RCT) compared to standard care [38]. For the intervention, the review presented high certainty of evidence for reduced complications and a slight reduction in mortality of up to 6 months, but no reduction in 30-day mortality (moderate certainty evidence) and low certainty of evidence relating to slight improvements in nutritional intake or status and weight or BMI, but no reduction in length of stay. The authors of the review call for standardised and more detailed reporting of dietary counselling methods, including frequency of input and ONS adherence [38]. 

### 3.3. Studies Employing Individualised Nutritional Care in Specific Patient Groups

Included in the review conducted by Wong et al. was the large multi-centre RCT (n = 2028) that showed early (within 48 h of admission) individualised nutritional support during hospital stay in a heterogeneous population of medical inpatients, increased daily energy and protein intake, lowered adverse clinical outcomes, improved survival and functional decline at 30 days and improved QOL [39]. The intervention included an individualised assessment of energy and protein goals; individualised nutrition care plans with the provision of nutrition support strategies, including dietary counselling and ONS if indicated; and on discharge, a protein target of 1.2–1.5 g/kg body weight/day. The lack of a continued effect at six months (primary outcome: mortality; secondary outcomes: re-admissions, falls and fractures, activities of daily living, QOL) may be related to the short, in-hospital, intervention period with only 24.2% of patients in the intervention group receiving ONS post-discharge [40]. 

In the secondary analyses of this trial, the INC was observed to be cost-effective as measured by a reduced risk of ICU admissions and complications [41]. In specific subgroups of patients, the INC resulted in a >50% reduction in the risk of 30-day mortality and improvements in functional and QOL outcomes in patients with age-related vulnerability [42], a 25% reduction in 30-day mortality in patients with lower respiratory tract infection [43], reduced risk of mortality and major cardiovascular events in patients with heart failure [44] and reduced risk of mortality and improved functional and quality of life outcomes in cancer patients at increased nutritional risk [45].

A systematic review on patients with head and neck cancer receiving radiotherapy (5 RCTs, n = 500) found that INC improved quality of life, nutritional status, treatment interruptions, symptom morbidity and nutritional intake. Weekly, individualised nutritional counselling was suggested as the optimal frequency, enabling dietitians to establish a therapeutic alliance with patients [46]. In the review, descriptions of INC differed between studies as follows: INC to meet patient-specific goals, tailored sample meal plans, recipe suggestions and information to minimise the side effects of the tumour and therapy, ONS if required [47,48]; behaviour-change counselling methods, including motivational interviewing and cognitive-behavioural therapy [49]; dietary counselling individualised to dietary requirements, usual intake, eating patterns and preferences [50]; and the latter combined with PEG feeding [51]. INC in all five RCTs was dietitian-led. 

An agreement of what constitutes INC, with better descriptions of INC in studies, is needed to facilitate comparisons and meaningfully incorporate this into nutrition guidelines to enable HCPs to implement INC in practice. 

## 4. Delivering Individualised Nutritional Care in Practice

INC can be used for patients in all healthcare settings, from those requiring first-line oral nutritional care, e.g., nutrient-dense meals, drinks and snacks, assistance with eating, monitoring intake and body weight, to complex nutritional care, e.g., dietary adjustment to manage the medical condition in combination with both enteral tube feeding and ONS. 

HCPs may be aware of the benefits of individualising care, but universal agreement on the definition and practical support on how to deliver INC in everyday clinical practice is needed. A practical model that helps keep what matters to patients as a priority could help nutrition experts and non-nutrition experts implement INC into everyday clinical practice.

### 4.1. The Nutrition Care Process—A Practical Model to Individualise Nutritional Care

The nutrition care process (NCP) is a systematic process used by healthcare professionals (predominantly dietitians) to deliver good nutritional care [35,52,53]. It demonstrates how professional knowledge and skills are integrated into evidence-based decision making to provide a consistent high quality of care [54]. The intention of the NCP is to put patients at the centre of good nutritional care [54] and help healthcare professionals demonstrate the effectiveness and value of nutritional care through a focus on outcome management [15,55]. It can be used in individual and population care delivery across a variety of different settings [56]. The NCP has primarily been developed and implemented by nutrition and dietetic professionals. AND adopted the NCP for use in the United States in 2003 [52], updated in 2017 [56], with nutrition care process terminology (NCPT) published in 2019 [57]. It has been widely adopted or adapted for implementation internationally [58] and may be known under different titles, such as the ‘Model and Process for Nutrition and Dietetic Practice’ implemented by the Association of UK Dietitians in the UK [54,59].

Although different adaptations of the NCP published by different organisations vary in the number of steps and in the terminology they employ, they can be consolidated as shown in Figure 3. Screening for risk of malnutrition is a step that precedes the NCP and prompts referral, where the NCP is subsequently employed. 

Care processes are not distinct to dietetics, other disciplines within healthcare also use similar systematic schemes to enhance patient-centred, outcome-focused, evidence-based care, e.g., the diagnostic process [60] and the pharmacists’ patient care process [61]. All healthcare professionals involved in the delivery of nutritional care should use a care process specific to nutrition. ESPEN state that ‘nutritional care should be provided in a systematic sequence that involves distinct interrelated steps and this systematic sequence is called the ‘nutrition care process’ [53]. In the ESPEN guidelines on clinical nutrition and hydration in geriatrics, Volkert et al. 2019 recommend that ‘a positive malnutrition screening shall be followed by systematic assessment, individualised intervention, monitoring and corresponding adjustment of interventions’ illustrating the embedding of the nutrition care process in key guidance for older persons [35]. 

The NCP has four interconnected steps divided into two core components: problem identification and problem solving. Problem identification includes: Step 1. Nutrition assessment and reassessment where HCPs explore patient’s experiences, needs, preferences and values and collect and interpret relevant anthropometric, biochemical, clinical, dietary, environmental, economic and functional data, and Step 2. Nutrition issue or diagnosis where nutritional problems that impact physical, mental and social wellbeing are identified, along with the causes, and signs and symptoms to phenotype patients. In the problem-solving phase HCPs can work collaboratively with patients to ensure a thorough understanding of the impact of the medical condition and nutritional diagnosis to motivate individual change, co-create personally relevant goals with patients and put in place a nutritional care plan, which includes defining nutritional targets as part of implementation (Step 3). In Step 4. The focus is on selecting appropriate outcome indicators and ensuring the timely monitoring of progress towards the agreed goals and resolution of the nutrition diagnosis where that is possible. Selecting outcome indicators that are relevant to co-created patient goals may help patients engage in ownership of their nutritional care plan and support self-monitoring and continuity of care across care settings.

The NCP is dynamic and cyclical; steps are revisited as new information is collected to update diagnoses, adjust interventions and modify goals. 

The NCP is at different stages of implementation in different healthcare systems and countries. Efforts are being made to define and standardise terminology associated with the NCP and to establish systems to collect and aggregate outcome data. However, even when the NCP is not fully integrated into healthcare systems, consideration of the four key steps by nutrition experts and non-nutrition experts during any interaction with patients who are malnourished or at risk of malnutrition, may help to guide efforts to ensure that nutritional care is individualised to patients’ needs, preferences and goals. 

Not confined to DRM, the NCP is used by dietitians across many different disease areas. Employing the NCP supports the use of individualised approaches, such as cognitive behavioural therapy and motivational interviewing, to target behaviour change. Positive effects of such techniques in the management of DRM in adults is evident in a study of their use in head and neck cancer patients [49].

### 4.2. An Interdisciplinary Approach to Individualised Nutritional Care

In a multidisciplinary care model, each professional utilises their own expertise to develop care goals independently with patients. In an interdisciplinary team approach, a collaborative care plan is developed to achieve common shared intervention goals. For example, the Systematised, Interdisciplinary Malnutrition Program for impLementation and Evaluation (SIMPLE) developed for the nutritional care of older patients with fragility fracture [62] mirrors the NCP and allows for locally adapted interventions. This approach has been shown to increase the proportion of patients who receive nutritional care and who report improved nutrition experiences, without the need for an increase in funding or dietetic resources, thus delivering value from the perspectives of what matters to patients and distribution of constrained dietetic resources [63]. The SIMPLE approach advocates the need for an open and honest discussion with patients about a malnutrition diagnosis to facilitate a shared decision-making approach to nutrition intervention planning [62]. Tools, such as SIMPLE and the Malnutrition Pathway [64], can incorporate exploring what matters to patients, shared decision making, evidence-based, multi-modal nutritional care interventions, goal setting and outcome monitoring to help individualise nutritional care delivered by non-nutrition expert health and care professionals in less complex cases. 

A scoping review of multidisciplinary provision of food and nutritional care to hospitalised adult in-patients explored the attributes that affect collaborative working in delivering patients’ nutritional care [65]. The review found that a wide range of healthcare disciplines and patients, relatives, volunteers are working collaboratively in delivering nutritional care, but that no studies specifically addressed the roles of collaborating professionals or their relationship to nutritional care processes. Features that supported collaborative nutritional care included role clarity, but with shared responsibility to allow for some interchangeability in roles to cover gaps or shift patterns, multidisciplinary relationships facilitated by effective communication, knowledge, shared learning and information sharing. The benefits of leadership support were not confined to management roles, but also include certain roles, such as nutrition advocates or champions to support collaborative, peer group change [65]. 

Further research is needed to clarify the optimal features and components of an interdisciplinary approach to INC, what factors facilitate or inhibit such an approach and what effect it could have on outcomes that matter to patients.

### 4.3. Measuring What Matters: The Co-Creation of Care

Co-creation of care is about shared understandings in relation to patient-centred communication and patient-centred goal setting - shared goals, shared knowledge and mutual respect. Co-creation is especially important in situations characterised by complex tasks, uncertainty and time constraints. Difficult and complex issues relevant to cases of DRM with multi-morbidity make the co-creation of care potentially valuable and may deliver better outcomes among these patients [3].

Identifying patient-centred goals and what matters to patients can help inform the selection of relevant outcome indicators. Differences may exist in outcomes that providers, commissioners or payors expect to be measured versus outcomes that may be meaningful to patients. The use of realistic, pragmatic, easy to use outcome parameters, e.g., symptoms, QOL, activities of daily living or fatigue scores, needs to be encouraged. Body weight and nutritional intake have a role but may be relied upon too heavily as outcome measures and may not be understood by patients as directly relevant to what matters to them in their day to day lives, such as the ability to participate in family life, undertake usual activities of daily living, participate in hobbies or stay as mobile as possible. Bridging the gap between a malnutrition diagnosis; outcomes, such as nutritional intake and body weight; and how these impact on overall health outcomes, such as survivorship or function, may help patients to better understand the potential impact of nutrition intervention and support engagement and self-management. Patient goals and the HCP’s goals for care may be similar, but may need to be phrased differently. A dual approach in research and practice may be needed, matching HCP goals and patient goals to make them explicit and meaningful. 

Continuity in outcome monitoring is needed, i.e., during intervention to support adherence and inform adjustment of intervention, but also at the end of intervention to demonstrate effectiveness. An interdisciplinary team approach to measuring outcomes could be taken, with data collated to inform and attract further investment in delivering interdisciplinary integrated care. Outcome monitoring could support continuity of care across healthcare settings, e.g., communication about nutritional care during key transition points in care. Opportunities for patients to self-monitor certain outcomes and to self-advocate for continuity of care warrants attention. 

## 5. Recommendations for Future Research

○Develop a consensus definition for INC as suggested in this review;○In trials and nutrition guidelines, emphasise shared decision making and focusing on what matters to patients to encourage consideration of these key elements of individualised nutritional care in practice;○Investigate the acceptability of the term and definition of INC to HCPs and patients; ○Increase the focus on outcomes that are meaningful to patients in trials;○Investigate whether individualising nutritional care leads to better outcomes in different patient groups in different care settings compared to standard non-INC;○Focus on models of care where evidenced-based nutritional care strategies are selected based on individual patient needs, preferences and goals rather than on pre-defined standard protocols where there is a hierarchy or stepwise approach to escalating nutrition intervention without any element of tailoring to patients’ specific needs;○Investigate the optimal methods of delivery of INC—face to face, telehealth, patient-preferred methods of providing education: websites, videos, printed materials, etc., cultural preferences; ○Develop evidence-based decision aids to use with patients to assist with shared decision making for INC;○Explore optimum language and ways to explain to patients the link between nutrition diagnosis, e.g., an acceptable lay term for malnutrition and its sequelae, and outcome to actively engage patients in nutritional care and support adherence;○Develop easy to use, practical tools or resources to enable patients to self-monitor and self-advocate for nutritional care and resources for HCPs to deliver these;○Investigate the impact of INC on patients with multi-morbidity. 

## 6. Conclusions

INC can be defined as nutritional care that is tailored to patients’ specific needs, preferences and goals, and includes certain key pillars, such as what matters to patients, shared decision making, evidence-informed multi-modal nutritional care and monitoring outcomes. There is evidence to support the use of INC in practice, but there is a need to further investigate the delivery and outcomes achievable in different patient groups in different care settings. There is also a need to better describe how nutritional care is individualised and how, by highlighting shared decision making, nutrition guidelines can help embed this concept into practice. The NCP offers a practical model for delivery of INC in clinical practice across all healthcare settings by all healthcare disciplines and could help healthcare professionals identify and promote the measurement of outcomes that matter to patients. 

## Figures and Tables

**Figure 1 nutrients-14-03534-f001:**
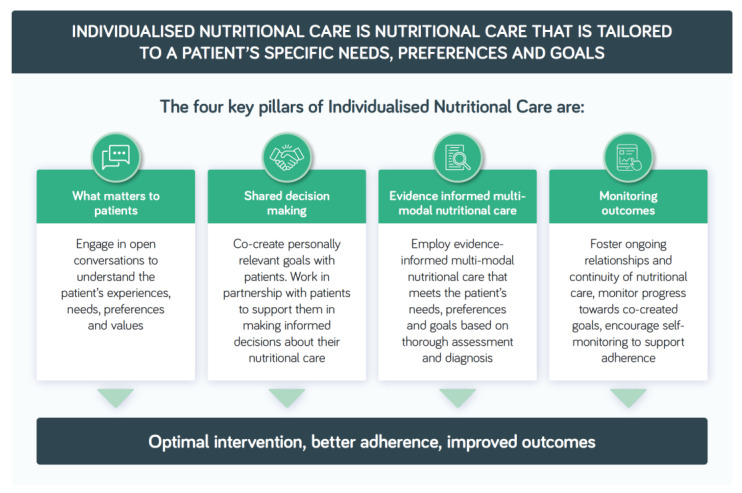
Individualised nutritional care in the management of disease-related malnutrition in adults.

**Figure 2 nutrients-14-03534-f002:**
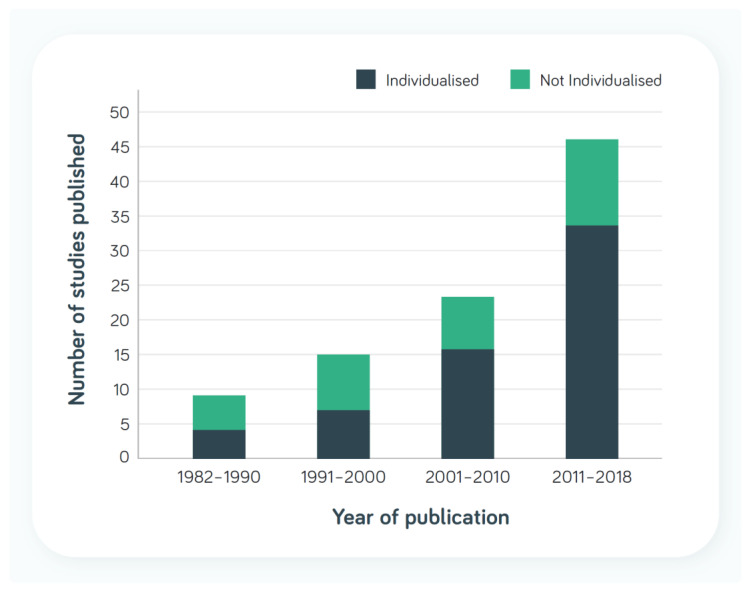
The number of studies reporting individualised or not-individualised interventions included in the Baldwin et al. review [5].

**Figure 3 nutrients-14-03534-f003:**
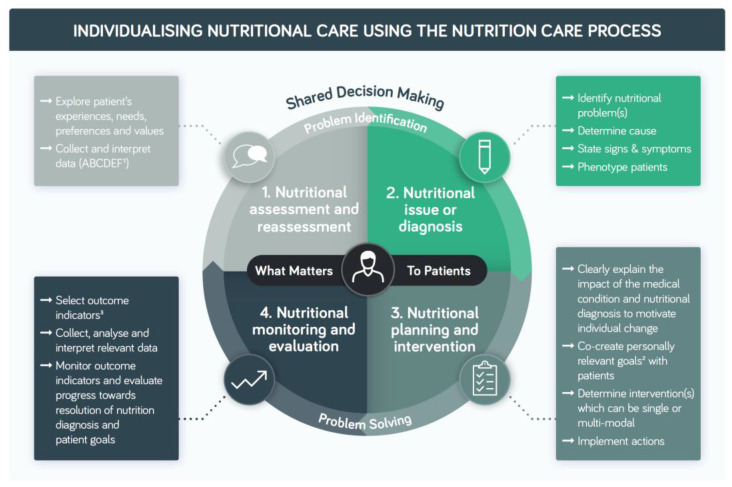
Individualising nutritional care in the management of disease-related malnutrition in adults using the nutrition care process (based on [53,54,56,59]). ^1^ Anthropometric, biochemical, clinical, dietary, environmental, economic, functional. ^2^ Goal = a measurable short-term aim set to be achieved by the next consultation or episode of care. ^3^ Outcome indicator = outcome indicator means a variable, parameter or tool that measures a change in status relating to the desired results of nutritional care.

**Table 1 nutrients-14-03534-t001:** Definitions of commonly used terms in patient-centred care.

Term	Definition	Comment
Needs	Wants that are essential, felt or expressed by the individual, rather than normative needs as defined by experts and compared against standards [11]	This should not be taken to mean that needs expressed or felt by patients cannot be measured
Preferences	An individual’s expression of desirability of one course of action, outcome or selection in contrast to others [12]	Patient preferences can be context specific, whereas patient values are not generally context specific Preferences are a consequence of values, and values are expressed through preferences
Values	A person’s beliefs or expectations about what is right or wrong. Values are latent traits [13]	
Goals	The end result or objective, which may be specified in advance [14]. It can also refer to a measurable short-term aim set to be achieved by the next consultation or episode of care [15]	Ideally, co-create goals with patients

**Table 2 nutrients-14-03534-t002:** Individualised nutritional care in nutrition guidelines.

Topic, Reference	Organisation, Resource Type, Year	Summary or Recommendations Relating to Aspects of INC ^1^
**General: relating to provision of nutrition support relevant to multiple patient groups**		
Home enteral nutrition [20]	ESPEN Practical Guideline 2022	Patient preference should be taken into account during decisions about method of home-enteral-nutrition administration
Hospital nutrition [21]	ESPEN Guideline 2021	Nutritional needs should be assessed individually Patient preferences, abilities, perspectives, religious beliefs and needs should be taken into account for hospital food delivery Food delivery is part of individualisation of nutritional care A combination of a specifically designed high-energy high-protein diet, snacking, ONS and nutritional counselling should be available in the acute hospital setting to provide the most individualised nutrition therapy
Ethical aspects of artificially administered nutrition and hydration [22]	ASPEN Position Paper 2021	Apply equally the four ethical principles of autonomy, beneficence, nonmaleficence and justice ^2^ Respect cultural values and religious beliefs For patients with cancer use patient-centred communication style that incorporates shared decision making For patients at the end of life, respect patient preferences and QOL goals with acceptance or refusal of nutritional care
Home parenteral nutrition (HPN) [23]	ESPEN Guideline 2020	Home parenteral nutrition (HPN) programmes shall provide individualised, safe, effective and appropriate nutrition support upon discharge from hospital A formal individualised HPN training programme for the patient and caregiver and home-care nurses shall be performed
Selection and care of central venous access devices for adult home parenteral nutrition administration [24]	ASPEN Guideline 2019	Acknowledges that the guideline covers a generalised outpatient population, but that infusion therapy selected should be tailored to the individual patient
Nutrition support: adult hospitalised patients [25]	ASPEN Standard 2018	The nutrition care plan should incorporate the wishes of patients and or caregivers Selection of venous access site should include consideration of patient preferences Selection of venous access device should include consideration of needs and goals (note the guideline does not specify if these are the patient’s needs and goals or as assessed by a healthcare professional (HCP))
Safe practices for enteral nutrition therapy [26]	ASPEN Consensus Recommendations2017	Choice of method of administration (bolus, intermittent, continuous) includes consideration of patient needs and goals (not specified if these are patient’s or as assessed by HCP) Choice of feed rate or duration includes consideration of patient lifestyle, goals and convenience For transition from enteral nutrition to oral feeding provide an individualised diet involving patient and family in food and oral supplement preferences Recommendations not intended to supersede judgement of HCP of individual patient circumstances
**Intensive care**		
Provision of nutrition support therapy in the adult critically ill patient [27]	ASPEN Guideline 2021	Clinicians should individualise protein prescriptions based on clinician judgment of estimated needs, until more data are available on the impact of higher protein with equivalent energy on outcomes
Clinical nutrition in the intensive care unit [28]	ESPEN Guideline 2019	Acknowledgement that the recommendations are a basis to support individualisation of nutritional care
**COVID-19**		
Nutritional management of individuals with obesity and COVID-19: ESPEN [29]	ESPEN Expert Statements and Practical Guidance 2021	Prevention of SARS-CoV-2 infection and poor COVID-19 outcomes: individualised recommendations should be provided as much as possible (to patients) with regard to exercise type, frequency, intensity and duration, by experienced healthcare professionals Nutritional treatment should continue after discharge from the hospital with individualised nutritional plans For intubated patients: mobilisation and physical activity should be implemented with individualised protocols During recovery: individualised exercise and physical activity programmes recommended
Nutritional management of individuals with SARS-CoV-2 infection [30]	ESPEN Expert Statements and Practical Guidance 2020	Measure energy needs using indirect calorimetry or weight-based formulae adjusted individually to account for nutritional status, physical activity, disease status and tolerance Estimate protein needs and adjust on an individual basis Nutritional treatment should continue after hospital discharge with ONS and individualised nutritional plans
Nutrition management for critically and acutely unwell hospitalised patients with coronavirus disease 2019 (COVID-19) in Australia and New Zealand [31]	Guideline 2020	Tailor nutritional care to pandemic capacity using an algorithm for initiating early enteral tube feeding in lower-nutritional-risk patients and individualised care for high-nutritional-risk patients where capacity allows
**Cancer**		
Clinical nutrition in cancer [32]	ESPEN Practical Guideline 2021	All cancer patients: individualised resistance exercise in addition to aerobic exercise to maintain muscle strength and muscle mass During radiotherapy: adequate nutritional intake should be ensured primarily by individualised nutritional counselling with or without with use of ONS
Cancer cachexia in adult patients [33]	ESMO Clinical Practice Guidelines 2021	Individualised nutritional intervention by a nutritionally trained professional team, alleviation and treatment of nutrition impact symptoms, psychological and social support, (supervised) physical exercise (strength, endurance), consider anticancer treatment Anti-cachexia treatment options: prioritise multimodal care Nutritional support and physiotherapy may be offered on an individual basis while carefully monitoring individual goals and QOL
Cancer-related malnutrition and sarcopenia [34]	COSA Position Statement 2020	All people with cancer-related malnutrition and sarcopenia should have access to the core components of treatment, including individualised medical nutrition therapy, targeted exercise prescription and physical activity advice and physical and psychological symptom management Treatment for cancer-related malnutrition and sarcopenia should be individualised, in collaboration with the multidisciplinary team (MDT)
**Older people**		
Clinical nutrition and hydration in geriatrics [35]	ESPEN Guideline 2019	Respecting the patient’s will and preferences is of utmost priority Values for energy and protein intake should be individually adjusted A positive malnutrition screening shall be followed by systematic assessment, individualised intervention, monitoring and corresponding adjustment of interventions Nutritional and hydration care for older persons shall be individualised and comprehensive in order to ensure adequate nutritional intake, maintain or improve nutritional status and improve clinical course and quality of life (QOL) Individualised nutritional counselling recommended Nutritional interventions in geriatric patients after hip fracture and orthopaedic surgery shall be part of an individually tailored, multidimensional and multidisciplinary team intervention in order to ensure adequate dietary intake, improve clinical outcomes and maintain quality of life
Individualised nutrition approaches for older adults: long-term care, post-acute care and other settings [36]	The AND Position Paper 2018	As part of the interprofessional team, registered dietitian nutritionists assess, evaluate and recommend appropriate nutrition interventions according to each individual’s medical condition, desires and rights to make healthcare choices

^1^ See Appendix A online for a table showing the information, recommendations or statements relevant to INC extracted from the original source publications from which this table is derived. ^2^ (1) Autonomy: respect the patient’s healthcare preferences; (2) beneficence: provide healthcare in the best interest of the patient; (3) nonmaleficence: do no harm and (4) justice: provide all individuals a fair and appropriate distribution of healthcare resources. ESPEN: European Society for Clinical Nutrition and Metabolism. ASPEN: American Society for Parenteral and Enteral Nutrition. COSA: The Clinical Oncology Society of Australia. ESMO: The European Society for Medical Oncology. AND: Academy of Nutrition and Dietetics.

**Table 3 nutrients-14-03534-t003:** Number of studies included by Baldwin et al. (2021) that describe individualised versus not-individualised nutritional care for management of disease-related malnutrition [5].

Comparison	Individualised	Not Individualised
DA vs. No DA	15	9
2. DA vs. ONS	2	10
3. DA + ONS vs. DA	6 ^1^	16
4. DA + ONS if required vs. No DA	28	3
5. DA + ONS vs. No DA + No ONS	12 ^2^	1
**Total ^3^**	**63**	**39**

^1^ In these six studies, nutritional counselling is given to reach or increase individualised energy and protein goals. ^2^ Intervention is tailored to individuals’ habitual intake or preferences. ^3^ The total number of studies is greater than 94 since some studies include comparisons in two or more parts of the review. ONS: oral nutritional supplements; DA vs. No DA: dietary advice versus no dietary advice; DA vs. ONS: dietary advice versus ONS; DA + ONS vs. DA: dietary advice plus ONS versus dietary advice; DA + ONS if required vs. No DA: dietary advice plus ONS if required versus no dietary advice; DA + ONS vs. No DA + No ONS: dietary advice plus ONS versus no dietary advice and no ONS.

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
