# Peer review of "Individualised Nutritional Care for Disease-Related Malnutrition: Improving Outcomes by Focusing on What Matters to Patients"

_nutrients, 2022, doi:10.3390/nu14173534_

Round 1

Reviewer 1 Report

The authors aim at reviewing the existing literature to collect the available recommendations on how to deliver individualized nutritional care (INC). The topic is of interest, and the manuscript highlights that the literature agrees on the clinical relevance of INC yet there is paucity of information on how to deliver it. The authors should address the following issues.

Major issues

1. The authors correctly point out that international guidelines use different words to define the same clinical process, or even vaguely describe INC. However, it is important that the authors implement the rigorous wording also in their manuscript. As an example, line 221 should report whether the intervention improves QoL; stating that the intervention "may lead to ..." add ambiguity to the phrase.

2. It is opinion of this reviewer that the authors should discuss whether INC comprises 2 major actions: 1) defining nutritional targets and 2) implementing them in the patients. Also, it should be clearly outlined that nutritional care should be considered a therapy and not just supportive care. In this respect, this reviewer fully support the statement by the authors that the efficacy of INC is the improvement of clinical outcomes rather than nutritional targets.

3. The authors appear to have overlook Bargetzi L et al. Ann Oncol 2021, which could be of further interest since it challenges the currently recommended nutritional targets.

4. This reviewer is skeptical about issuing recommendations on how nutritional vare can be individualised. Considering that patients are by definition different from each other, the ability to individualize treatments should have been learnt during the professional careers. As an example, there is no guideline for individualized anti-hypertensive treatment, but HCPs use the drug which best fit with the clinical history and patient's needs.

Author Response

Response to reviewer 1

Reviewer’s feedback: ‘The authors aim at reviewing the existing literature to collect the available recommendations on how to deliver individualized nutritional care (INC). The topic is of interest, and the manuscript highlights that the literature agrees on the clinical relevance of INC yet there is paucity of information on how to deliver it. The authors should address the following issues.’

Authors’ response: We thank the reviewer for their careful consideration of our paper and for the helpful feedback which we address point by point below. We are pleased that the reviewer considers that the topic is of interest.

  1. Reviewer’s comment: ‘The authors correctly point out that international guidelines use different words to define the same clinical process, or even vaguely describe INC. However, it is important that the authors implement the rigorous wording also in their manuscript. As an example, line 221 should report whether the intervention improves QoL; stating that the intervention "may lead to ..." add ambiguity to the phrase.’

Authors’ response: Thank you, this is an important observation and one that we considered very carefully whilst drafting the manuscript. In particular, we discussed wording such as that used at line 221, namely ‘..DA+ONS may lead to fewer complications and shorter length of hospital stay after 3 months in adult patients with DRM’ to decide whether it was appropriate to use the words ‘may lead’.  Although this terminology could be seen to be vague, it is the terminology recommended for use with the Grading of Recommendations Assessment, Development and Evaluation (GRADE) approach used by Baldwin et al., and outlined in The Cochrane Handbook for Systematic Reviews of Interventions, (see Chapter 15, Table 15.6.b Suggested narrative statements for phrasing conclusions[i]). For this reason, we retained the terminology ‘may lead’ to ensure that we accurately reflected the results found by Baldwin et al. To make the fact that this was the result reported by Baldwin et al. clearer, the manuscript has been amended to read as follows at line 251: ‘Additionally, Baldwin et al. reported that dietary advice plus ONS may lead to fewer complications and shorter length of hospital stay after 3 months in adult patients with DRM [5].’

  1. Reviewer’s comment: ‘It is opinion of this reviewer that the authors should discuss whether INC comprises 2 major actions: 1) defining nutritional targets and 2) implementing them in the patients. Also, it should be clearly outlined that nutritional care should be considered a therapy and not just supportive care. In this respect, this reviewer fully supports the statement by the authors that the efficacy of INC is the improvement of clinical outcomes rather than nutritional targets.’

Authors’ response: Thank you, we agree that INC also includes the two major actions outlined by the reviewer, namely 1) defining nutritional targets and 2) implementing them in the patients.  We believe that defining nutritional targets and implementing them in patients is part of delivery of INC for which we highlight the use of the Nutrition Care Process (Section 4. Delivering INC in practice). Both of these actions fall under step 3 ‘Nutritional planning and intervention’. To ensure that these two actions are clearer to readers we have amended lines 416 to 241 to read as follows: ‘In the problem solving phase HCPs can work collaboratively with patients to ensure a thorough understanding of the impact of the medical condition and nutritional diagnosis to motivate individual change, co-create personally relevant goals with patients and put in place a nutritional care plan which includes defining nutritional targets as part of implementation (Step 3.).’

We agree that that nutritional care should be considered a therapy and not just supportive care and therefore to ensure that this point is clearly made we include it at line 64 as follows: ‘Nutritional care is a therapy and not just supportive care.’

  1. Reviewer’s comment: ‘The authors appear to have overlook Bargetzi L et al. Ann Oncol 2021, which could be of further interest since it challenges the currently recommended nutritional targets.’

Authors’ response: Thank you, this additional important paper has been included alongside the summary of results for other subgroup analyses of the EFFORT trial (see lines 341 to 348) as follows: ‘In secondary analyses of this trial, the INC was found to be cost-effective as measured by a reduced risk of ICU admission and complications [41]. In specific subgroups of patients, the INC resulted in a >50% reduction in the risk of 30-day mortality and improvements in functional and QOL outcomes in patients with age-related vulnerability [42], a 25% reduction in 30-day mortality in patients with lower respiratory tract infection [43], reduced risk of mortality and major cardiovascular events in patients with heart failure [44] and reduced risk of mortality and improved functional and quality of life outcomes in cancer patients at increased nutritional risk [45].

  1. Reviewer’s comment: ‘This reviewer is skeptical about issuing recommendations on how nutritional care can be individualised. Considering that patients are by definition different from each other, the ability to individualize treatments should have been learnt during the professional careers. As an example, there is no guideline for individualized anti-hypertensive treatment, but HCPs use the drug which best fit with the clinical history and patient's needs.’

Authors’ Response: Thank you for this helpful feedback on the recommendations for future research. Our intention in bullet point two was to convey that, in clinical trials and in expert guidelines, two of the key pillars of INC i.e., shared decision making and what matters to patients could be better emphasised to encourage consideration of these in practice. It was not the intention to suggest that expert guidelines give specific recommendations on nutritional care for each individual patient, since, as the reviewer rightly highlights, that is the skill of the HCP. To clarify this point, the wording of bullet point two has been changed as follows: ‘In trials and in nutrition guidelines, emphasise shared decision making and focusing on what matters to patients to encourage consideration of these key elements of individualised nutritional care in practice.’

[i] Schünemann HJ, Vist GE, Higgins JPT, Santesso N, Deeks JJ, Glasziou P, Akl EA, Guyatt GH. Chapter 15: Interpreting results and drawing conclusions. In: Higgins JPT, Thomas J, Chandler J, Cumpston M, Li T, Page MJ, Welch VA (editors). Cochrane Handbook for Systematic Reviews of Interventions version 6.3 (updated February 2022). Cochrane, 2022. Available from www.training.cochrane.org/handbook.

Reviewer 2 Report

I have enjoyed reviewing this important and very interesting review paper. It has been thoroughly and thoughtfully constructed and will be valuable to the healthcare profession community.   It contains an excellent summary of the literature in relation to individualised nutritional care and a detailed discussion of the concept and the direction of its development in the future. 

There are a number of places were the text could be clarified and there are various minor typos and grammatical errors. Please see attached word document, which I hope will serve to improve this excellent paper.

Line 87, 99, 102, 367, 446: patients’ not patient’s

Table1: Definition of ‘Needs’ – I think you need a comma after essential unless this is meant to read essentially felt?  The comment associated with ‘Goals’ reads like an instruction in contrast to the other comments which explain the item further; is this what the authors intended? Should it read Ideally co-create goals…..or something similar?

Line 111: this paragraph is about the problems of vague terms or the lack of understanding about terms. You use the term ‘humanistic behaviour’; it could be useful to explain what you mean by this. I am an experienced dietitian, but I had to look it up (Perhaps I am simply showing my ignorance!). A short definition in brackets after the term could help readers’ understanding.

2.2 Defining INC – It is better not to use abbreviations in titles. Please write INC in full.  See also all sub-headings throughout unless editor advises otherwise.

Figure 1: 1st box – what matters to patients – either move the apostrophe (patients’ not patient’s) or write….understand the patient’s experiences,…. Similarly for box 3 – Evidence informed care……meets the patient’s needs…..

See previous comment about abbreviations in titles – better to put INC in full in the figure title. Figures and tables should stand alone.

Line 142 – monitoring and evaluation – not /

Line 146: delete individual and put ‘the’

Table 2: 1st row – do you mean practical or practice guideline?

Row 3 – 1st bullet - four not 4; 4th bullet – avoid / as abbreviation for ‘or’

Last row – perhaps use ‘AND’ and put it in full in the footnote to be consistent with the rest of the table.

Line 187: avoid / as abbreviation for ‘and’ and ‘or’ – assessment and adjustment; INC plans or nutrition intervention etc

Line 191: I think this should be ….the patient’s personal goals…..

Line 213: end bracket ) not needed.

Line 213 to 219 – detail of comparisons – I don’t think this level of detail is required here and it is quite hard work to read it. You could say:

Comparisons were made between various combinations of dietary advice, with and without ONS, and no dietary advice. Overall, there was ………..

For the table 3 – which stand alone, you really should add the full text for all the abbreviations again. These could be in the table footnote rather than the manuscript text.

Line 305: this paragraph would read better if you add an example of complex nutritional care to balance the examples of first line oral nutritional care.

Line 310 delete ‘front of mind’ as replace with ‘as the priority’ – front of mind seems rather colloquial for a scientific paper

Line 321: I understood the Nutrition Care Process to be a single process which can be adapted rather than lots of different nutritional care processes. Indeed you start this paragraph with: The Nutrition Care Process is a ……..  In line 321 you say NCPs have been developed – should this not read  - The NCP has been primarily developed and implemented ….Line 322 - AND adopted a version of the NCP for use in the US……  Line 326 - In the UK the BDA has adopted a version called Model and Process…..etc. Perhaps review and/or reorder this and following paragraphs to ensure the narrative is consistent.

Line 347 – replace / with comma – environmental, economic, and ….  Nutrition Issue or Diagnosis

Figure 3 title – ‘ not / for environment, economic, functional….

Line 377 – you don’t need to include abbreviations here. They are not subsequently used.

Line 477 – I think you need to decide what you mean here A NCP or The NCP – delegating the choice to the reader is not helpful. See previous comments about line 321.

Author Response

Response to reviewer 2

Reviewer’s feedback: ‘I have enjoyed reviewing this important and very interesting review paper. It has been thoroughly and thoughtfully constructed and will be valuable to the healthcare profession community.   It contains an excellent summary of the literature in relation to individualised nutritional care and a detailed discussion of the concept and the direction of its development in the future.’

There are a number of places where the text could be clarified and there are various minor typos and grammatical errors. Please see attached word document, which I hope will serve to improve this excellent paper.

Authors’ response: We thank the reviewer for their careful consideration of our paper and for the helpful feedback which we address point by point below. We are very pleased that the reviewer considers the paper to be important, interesting, well-constructed and of interest to healthcare professionals.

  1. Reviewer’s comment: ‘Line 87, 99, 102, 367, 446: patients’ not patient’s’

Authors’ Response: Thank you, corrections made as requested.

  1. Reviewer’s comment: ‘Table1: Definition of ‘Needs’ – I think you need a comma after essential unless this is meant to read essentially felt? The comment associated with ‘Goals’ reads like an instruction in contrast to the other comments which explain the item further; is this what the authors intended? Should it read Ideally co-create goals…..or something similar?’

Authors’ Response: Thank you, the following amendments have been made to Table 1. A comma has been inserted after the word ‘essential’. ‘Co-create goals with patients’ has been amended to read ‘Ideally co-create goals with patients’.

  1. Reviewer’s comment: ‘Line 111: this paragraph is about the problems of vague terms or the lack of understanding about terms. You use the term ‘humanistic behaviour’; it could be useful to explain what you mean by this. I am an experienced dietitian, but I had to look it up (Perhaps I am simply showing my ignorance!). A short definition in brackets after the term could help readers’ understanding.’

Authors’ Response: Thank you, line 114 has been amended to include the explanation of ‘humanistic behaviours’ as described by Sladdin et al. 2017. The text now reads: ‘..humanistic behaviours (behaviours or characteristics perceived as helpful and motivating to patients)’.

  1. Reviewer’s comment: ‘2.2 Defining INC – It is better not to use abbreviations in titles. Please write INC in full. See also all sub-headings throughout unless editor advises otherwise.’

Authors’ Response: Thank you, this correction has been made throughout the manuscript.

  1. Reviewer’s comment: ‘Figure 1: 1st box – what matters to patients – either move the apostrophe (patients’ not patient’s) or write….understand the patient’s experiences,…. Similarly for box 3 – Evidence informed care……meets the patient’s needs…..’

Authors’ Response: Thank you, the text in box 1 and box 3 in Figure 1 have been amended as requested.

  • In box 1 the text now reads as ‘Engage in open conversations to understand the patient’s experiences, needs, preferences and values’.
  • In box 3 the text now reads as ‘Employ evidence-informed multi-modal nutritional care that meets the patient's needs, preferences and goals based on thorough assessment and diagnosis.’

Please note that the revised figure is currently being updated by the artwork agency and will be submitted as soon as available

  1. Reviewer’s comment: ‘See previous comment about abbreviations in titles – better to put INC in full in the figure title. Figures and tables should stand alone.’

Authors’ Response: Thank you, the abbreviation INC in the title of Figure 1 has been amended and is now written in full. All other table and figure titles in the manuscript have been checked and any abbreviations written in full.

  1. Reviewer’s comment: ‘Line 142 – monitoring and evaluation – not /’

Authors’ Response: Thank you, the oblique symbol has been replaced with the word ‘and’.

  1. Reviewer’s comment: ‘Line 146: delete individual and put ‘the’

Authors’ Response: Thank you, the word individual bas been replaced with the word ‘the’.

  1. Reviewer’s comment: ‘Table 2: 1st row – do you mean practical or practice guideline?’

Authors’ Response: Thank you, the word ‘practical’ is correct since the title of the guideline is ‘ESPEN practical guideline: Home enteral nutrition’

  1. Reviewer’s comment: ‘Row 3 – 1st bullet - four not 4; 4th bullet – avoid / as abbreviation for ‘or’

Authors’ Response: Thank you, row 3 now includes the word ‘four’ in place of the digit ‘4’. The 4th bullet has been amended to remove the oblique symbol and now reads as ‘For patients at the end-of-life, respect patient preferences and QOL goals with acceptance or refusal of nutritional care’.

  1. Reviewer’s comment: ‘Last row – perhaps use ‘AND’ and put it in full in the footnote to be consistent with the rest of the table.’

Authors’ Response: Thank you, ‘AND’ now replaces the words ‘Academy of Nutrition and Dietetics’ in the last row of Table 2 and the footnote has been updated as requested.

  1. Reviewer’s comment: ‘Line 187: avoid / as abbreviation for ‘and’ and ‘or’ – assessment and adjustment; INC plans or nutrition intervention etc.’

Authors’ Response: Thank you, at line 222 the oblique symbol has been removed and either the word ‘and’ or the word ‘or’ used, where appropriate, in its place. Please note that the full manuscript has been checked and further instances where the oblique symbol was used have also been amended where appropriate.

  1. Reviewer’s comment: ‘Line 191: I think this should be ….the patient’s personal goals…..’

Authors’ Response: Thank you, this has been corrected to read ‘…the patient’s personal goals…’

  1. Reviewer’s comment: ‘Line 213: end bracket ) not needed.’

Authors’ Response: Thank you, the end bracket has been removed.

  1. Reviewer’s comment: ‘Line 213 to 219 – detail of comparisons – I don’t think this level of detail is required here and it is quite hard work to read it. You could say: Comparisons were made between various combinations of dietary advice, with and without ONS, and no dietary advice. Overall, there was ………..’

Authors’ Response: Thank you, this is a very helpful comment and considerably simplifies this complex paragraph. The change suggested has been made and the text at lines 248 to 251 now reads as follows: ‘Comparisons were made between various combinations of dietary advice, with and without ONS, and no dietary advice. Overall, there was no difference in mortality for any of the comparisons, however positive changes were found for dietary intake, body weight, fat-free mass and QOL.’

  1. Reviewer’s comment: ‘For the table 3 – which stand alone, you really should add the full text for all the abbreviations again. These could be in the table footnote rather than the manuscript text.’

Authors’ Response: Thank you, the full text for the abbreviations for the various combinations of dietary advice with and without oral nutritional supplements included in the Baldwin review are now shown as a footnote to Table 3 as suggested.

  1. Reviewer’s comment: ‘Line 305: this paragraph would read better if you add an example of complex nutritional care to balance the examples of first line oral nutritional care.’

Authors’ Response: Thank you, we agree an example would be useful here and have therefore amended the text at lines 367 to 368 to read as follows: ‘INC can be used for patients in all healthcare settings, from those requiring first line oral nutritional care e.g., nutrient dense meals, drinks and snacks, assistance with eating, monitoring intake and body weight, to complex nutritional care e.g. dietary adjustment to manage the medical condition in combination with both enteral tube feeding and ONS.’

  1. Reviewer’s comment: ‘Line 310 delete ‘front of mind’ as replace with ‘as the priority’ – front of mind seems rather colloquial for a scientific paper’

Authors’ Response: Thank you, the expression ‘front of mind’ has been replaced with the phrase ‘as the priority’ at line 377.

  1. Reviewer’s comment: ‘Line 321: I understood the Nutrition Care Process to be a single process which can be adapted rather than lots of different nutritional care processes. Indeed, you start this paragraph with: The Nutrition Care Process is a …….. In line 321 you say NCPs have been developed – should this not read  - The NCP has been primarily developed and implemented ….Line 322 - AND adopted a version of the NCP for use in the US……  Line 326 - In the UK the BDA has adopted a version called Model and Process…..etc. Perhaps review and/or reorder this and following paragraphs to ensure the narrative is consistent.’

Authors’ Response: Thank you. Yes the Nutrition Care Process is a single process. To make this clearer the change suggested at line 321 (now line 388) has been made and now reads as follows: ‘The NCP has primarily been developed and implemented by nutrition and dietetic professionals.’ At line 389 the abbreviation ‘AND’ now replaces the words ‘Academy of Nutrition and Dietetics’ as requested.

Section 4.1 has been reviewed and re-ordered to make the narrative more consistent. The following changes have been made:

  • The Nutrition Care Process is now referred to throughout as ‘the’ nutrition care process, rather than a nutrition care process
  • The paragraph starting ‘Although different adaptations of the NCP published by different organisations vary in the number of steps….’ now appears as the second paragraph in this section rather than the third paragraph in this section.

  1. Reviewer’s comment: ‘Line 347 – replace / with comma – environmental, economic, and …. Nutrition Issue or Diagnosis’

Authors’ Response: Thank you, the changes requested have been implemented.

  1. Reviewer’s comment: ‘Figure 3 title – ‘ not / for environment, economic, functional….’

Authors’ Response: Thank you, the oblique symbol has been removed and replaced with a comma as requested. Please also note that in Figure 3 the oblique symbols have been removed within in the figure to reflect the changes to the text i.e.

  • Nutritional Assessment/Reassessment now appears as Nutritional Assessment and Reassessment
  • Nutritional issue/diagnosis now appears as Nutritional issue or diagnosis

Please note that the revised Figure is currently being updated by the artwork agency and will be submitted as soon as available

  1. Reviewer’s comment: ‘Line 377 – you don’t need to include abbreviations here. They are not subsequently used.’

Authors’ Response: Thank you, the abbreviations CBT and MI have been removed.

  1. Reviewer’s comment: ‘Line 477 – I think you need to decide what you mean here A NCP or The NCP – delegating the choice to the reader is not helpful. See previous comments about line 321.’

Authors’ Response: Thank you, the NCP is now referred to as ‘the’ NCP throughout.

Round 2

Reviewer 1 Report

This reviewer would like to thank the authors for taking care of the concerns raised by their previous draft of the manuscript. All concerns have been addressed adequately, no further action is required.